# Differential Primary Seed and Fruit Dispersal Mechanisms and Dispersal Biomechanics in Invasive Dehiscent and Indehiscent-Fruited *Lepidium* Species

**DOI:** 10.3390/plants14030446

**Published:** 2025-02-03

**Authors:** Said Mohammed, Tina Steinbrecher, Gerhard Leubner-Metzger, Klaus Mummenhoff

**Affiliations:** 1Department of Biology, College of Natural and Computational Sciences, Debre Berhan University, Debre Berhan 445, Ethiopia; 2Department of Biology, University of Osnabrück, Barbarastraße 11, D-49076 Osnabrück, Germany; kmummenh@uni-osnabrueck.de; 3Department of Biological Sciences, Royal Holloway University of London, Egham TW20 0EX, UK; tina.steinbrecher@rhul.ac.uk (T.S.); gerhard.leubner@rhul.ac.uk (G.L.-M.)

**Keywords:** dispersal biomechanics, distribution, *Lepidium*, primary dispersal, seed/fruit detachment

## Abstract

This study explores primary dispersal, which involves diaspores’ detachment directly from mature plants, and secondary dispersal, which encompasses any further dispersal occurring after the primary dispersal. A comparison of the primary dispersal vectors of the invasive dehiscent fruit producing *Lepidium campestre* and the indehiscent fruit producing *Lepidium draba* was conducted. These vectors were examined in relation to the native and introduced distribution ranges of the species, and regarding biomechanical forces required detaching the fruits from mature plants. Our findings reveal that rainfall and animal contact serve as primary dispersal vectors for *Lepidium campestre*, while animal contact is rarely involved in primary dispersal of *Lepidium draba*. Primary dispersal is more important for *Lepidium campestre* than *Lepidium draba*, which requires significantly greater force for fruit detachment. While previous studies indicate wind and rainfall as major secondary dispersal vectors for *Lepidium draba*, our results suggest secondary dispersal via mucilage on seeds is more crucial for this species. The strong reliance of *Lepidium draba* on secondary dispersal, enabling long-distance dispersal, could contribute to its invasion success, linking it to the species’ geographic distribution. Understanding these different dispersal strategies is essential for effective management of invasive species.

## 1. Introduction

Dispersal of seeds or fruits is a vital process in the population dynamics of plant species, shaping their potential for recruitment and influencing subsequent stages of growth [1,2]. A key advantage of dispersal is that it facilitates the colonization of unpredictable but favorable sites, which is particularly significant for invasive species [1,3,4,5]. Additionally, dispersal reduces the potential for competition between the parent and offspring, and among offspring [1]. The season of seed maturity [6,7], environmental factors [7,8], and the morphology of seeds and fruits [9] all play a role in determining the mode of dispersal. Diaspores can be dispersed by water [10], animals including humans [11,12], and/or wind [13,14]. The process of dispersal encompasses two main phases: primary and secondary dispersal [7,14]. Primary dispersal occurs when diaspores detach directly from the mother plant [1,2,7], while secondary dispersal refers to any further dispersal subsequent to primary dispersal [7,10,14]. In this study, we focus on the primary dispersal of the two invasive closely related species: *Lepidium draba*, which produces indehiscent fruits, and *Lepidium campestre*, which produces dehiscent fruits (Figure 1). *Lepidium draba* L. (syn. *Cardaria draba* (L.) Desvaux) is commonly known as heart-podded hoary cress ([15,16,17], Figure 1). The dehiscent fruit-producing *Lepidium campestre* (L.) (Linnaeus) W. T. Aiton is commonly known as field pepperwort [18]; Figure 1(iv–v). The INDEHISCENT (*IND*) gene is important for the formation of the separation and lignified layer that control the level of dehiscence of a fruit [19,20]. Downregulation of its expression in the transgenic *L. campestre RNAi-LcIND* line therefore resulted in an indehiscent fruit-producing *L. campestre* line [21]

The dispersal units of *Lepidium draba* consist of the whole indehiscent fruit, leading to germination directly out of the fruit [22,23,24,25]. In contrast, seeds released from the dehiscent fruits serve as the dispersal units in *Lepidium campestre* [25,26,27]. An interesting common characteristic of the studied species is the ability of the seeds to become mucilaginous upon wetting [23,28]; Figure 1. Furthermore, ref. [24] observed that freshly harvested fruits of *L. draba* exhibit pericarp-mediated chemical dormancy, attributed to the inhibitory effect of abscisic acid (ABA) present in the fresh pericarp tissues. Similarly, ref. [25] reported that *L. campestre* has physiological seed dormancy, a finding supported by Partzsch [26]. It is worth noting that after-ripening releases dormancy in both *L. draba* and *L. campestre* [24,25].

The secondary dispersal mechanisms of *L. draba* and *L. campestre* have been studied in context of the seed mucilage of the two species and the indehiscent fruits of *L. draba* [28]; Figure 1. In this case, wind and rainfall are the most probable potential secondary dispersal mechanisms in *L. draba*, as the seeds and fruits of this species are strongly dispersed by these vectors. On the other hand, the seeds of *L. campestre*, released from the dehiscent fruits, are more likely to adhere to animals as a means of secondary dispersal. However, the primary dispersal vectors of the seeds and fruits dispersed directly from the mother plant are largely unknown for the two species.

Hence, this study aims to compare the primary dispersal vectors of the two invasive *Lepidium* species, *L. draba* and *L. campestre*, and to provide additional insights into the dispersal mechanisms of these study species. We hypothesize that primary dispersal vectors such as rainfall and animal contacts could benefit *L. campestre*, given that the species produces dehiscent fruits that are easily detached on contact with animals and rainfall. The distribution patterns and invasive potential of the study species is discussed in the context of both primary and secondary dispersal mechanisms.

## 2. Results

### 2.1. Detachment of Seeds and Fruits from Replum and Pedicel Junction in Lepidium Species: Raindrop and Mechanical Contact Experiments

In the dehiscent fruit producing *Lepidium campestre*, seeds and fruit valves were detached together in both dry and wet dehiscent fruits (Figure 2A). The force required to detach seeds from the replum was then analyzed for dry M+ (dry mucilaginous seeds), wet M+ (wet mucilaginous seeds), and re-dried M+ (re-dried mucilaginous seeds) (Figure 2B). There was approximately a 50% chance of the fruit valves and seeds detaching together in dry dehiscent fruits (Figure 2A) and over 60% in wet dehiscent fruits (Figure 2A). Although the proportion of wet dehiscent fruits with seeds and fruit valves detaching together was higher, there was no statistically significant difference (*p* > 0.05) compared to dry dehiscent fruits of *L. campestre* (Figure 2A). Furthermore, the force required to detach seeds from the replum showed that re-dried M+ and dry M+ seeds needed more force compared to wet M+ seeds (Figure 2B). There was no significant difference between the force required for detaching dry M+ and re-dried M+ seeds in *L. campestre* (Figure 2B). However, both dry M+ seeds and re-dried M+ seeds required significantly more force to detach from the replum compared to wet M+ seeds (Figure 2B).

The raindrop impact analysis revealed that the indehiscent fruits of *L. draba* required a greater number of raindrops to detach fruits from the replum and pedicel compared to the re-dried M+, dry M+, and wet M+ seeds of *L. campestre* (Figure 2C). This indicates that the indehiscent fruits of *L. draba* exhibit stronger attachment to the replum and pedicel compared to the seeds of *L. campestre*. The significantly higher number of raindrops needed to detach *L. draba* fruits from the replum and pedicel indicate the raindrop impacts are less likely primary dispersal vectors in *L. draba*, but are important in seed dispersal in *L. campestre* (Figure 2C).

When comparing the different conditions for the mucilaginous *L. campestre* seeds, re-dried seeds strongly adhered to the replum and pedicel as compared to dry M+ and wet M+ seeds, and this difference was statistically significant (*p* < 0.001, Figure 2C). Additionally, the dry M+ seeds of *L. campestre* displayed stronger adherence to the replum and pedicel compared to the wet M+ seeds, and this difference was also statistically significant (*p* < 0.001). Overall, the raindrop impact analysis demonstrated that the indehiscent fruits of *L. draba* exhibit strong attachment to the pedicel and replum compared to the re-dried M+, dry M+, and wet M+ seeds of *L. campestre*. In terms of adherence to animals, the indehiscent fruits of *L. draba* showed the highest adherence, followed by the re-dried M+ seeds of *L. campestre*, followed by the dry M+ seeds of *L. campestre*, with the least adherence observed in the wet M+ seeds of *L. campestre* (Figure 2C).

The raindrop impact experiment was complemented by the analysis of the number of animal (sheep wool) contact required to detach seeds and fruits from the replum and pedicel. The results showed that the indehiscent fruits of *L. draba* required the highest number of mechanical contact to detach fruits from the replum and pedicel compared to the seeds of *L. campestre* (Figure 2D). There was a statistically significant difference (*p* < 0.0001) between the indehiscent e fruits of *L. draba* and the seeds (dry M+, wet M+, and re-dried M+ seeds) of *L. campestre* (Figure 2D). Subsequently, it was observed that the re-dried M+ seeds of *L. campestre* required a relatively high amount of contact to detach seeds from the replum and pedicle compared to dry M+ and wet M+ seeds of the same species. Significant differences (*p* < 0.001) were also observed among the seeds (dry M+, re-dried M+, and wet M+ seeds) of *L. campestre*, indicating that the re-dried M+ seeds showed stronger adherence (*p* < 0.001), followed by the wet M+ seeds (*p* < 0.001), and the least adherence was observed in the dry M+ seeds (*p* < 0.001) in *L. campestre* (Figure 2D). The number of mechanical contacts needed to detach fruits and seeds from the replum and pedicel junction revealed that the highest number of contact was required for the indehiscent fruits of *L. draba*, followed by the re-dried M+ seeds of *L. campestre*, and then the wet M+ seeds of *L. campestre*. The least number of mechanical contacts required to detach seeds from the replum and pedicel junction was observed in the dry M+ seeds of *L. campestre* (Figure 2D). Statistically significant differences (*p* < 0.001) were reported between the indehiscent fruits of *L. draba* and the seeds of the dehiscent fruit producing *L. campestre* (Figure 2D).

### 2.2. Comparative Analysis of Seed and Fruit Detachment from the Replum and Pedicel Junction in Lepidium Species Under Simulated Rainfall, Wind, and Mechanical Contacts

The experiment assessing diaspore attachment to sheep wool revealed that none of the indehiscent fruits of *L. draba* and none of the dry M+ seeds of *L.campestre* attached to sheep wool (re Figure 3A). However, both the wet M+ and re-dried M+ seeds of *L. campestre* showed attachment to sheep wool. Specifically, over 40% of the wet M+ seeds and less than 10% of the re-dried M+ seeds of *L. campestre* attached to sheep wool (Figure 3A). Therefore, the wet M+ seeds of *L. campestre* demonstrated the highest attachment to sheep wool (*p* < 0.001), followed by the re-dried M+ seeds of *L. campestre*. These results indicate that the dehiscent fruit producing *L. campestre* has a high potential attaching to sheep wool compared to the indehiscent fruit producing *L. draba* (Figure 3A).

The analysis of diaspore detachment in mechanical contacts revealed that all of the diaspores of the dehiscent fruit producing *L. campestre* were detached by mechanical contacts (Figure 3B). However, the diaspores of the indehiscent fruit producing *L. draba* showed that most of the fruits remained attached to the infructescence after mechanical contacts (Figure 3B). Specifically, the wet indehiscent fruits (>90%) of *L. draba* exhibited the highest number of fruits that remained on the infructescence, followed by dry indehiscent fruits (70%) of *L. draba* (Figure 3B). There were significant differences (*p* < 0.001) between the wet indehiscent and dry indehiscent fruits of *L. draba* and the seeds of *L. campestre* (Figure 3B). This result indicates that the indehiscent fruits producing *L. draba* fruits remain attached to the mother plant after potential mechanical contacts, whereas seeds of *L. campestre* fall to the ground easily when there is mechanical contact with the diaspores (Figure 3B).

The diaspores’ detachment through the rainfall experiment showed that none of the wet and dry indehiscent fruits were detached from the replum and pedicel after the required rainfall experiments was carried out (Figure 4A). This suggests that rainfall has no probability to detach fruits of *L. draba* from the mother plant. Furthermore, the re-dried M+ seeds of *L. campestre* displayed more resistance, with approximately 80% of the seeds remaining attached to the mother plant after rainfall simulation experiments (Figure 4A). In comparison, the wet M+ seeds exhibited approximately 23% remaining on the mother plant, and the dry M+ seeds of *L. campestre* showed 20% of the diaspores remaining after the rainfall simulation experiment (Figure 4A).

The indehiscent fruits of *L. draba* showed the highest resistance to detachment (less detachment, *p* < 0.0001), followed by the re-dried M+ seeds of *L. campestre* (*p* < 0.0001), and then the wet M+ and dry M+ seeds of *Lepidium campestre* (*p* < 0.001) (see also Figure 4A). Therefore, rainfall might not be a primary dispersal vector for the indehiscent *L. draba* fruits, but it could be a primary dispersal vector for the dehiscent fruit producing *L. campestre* seeds (Figure 4A).

The wind experiment to detach diaspores from the replum and pedicle indicated that wind is less likely to be the primary dispersal vector in *L. draba* and *L. campestre* (Figure 4B). None of the indehiscent fruits of *L. draba* detached, and approximately 98% of the re-dried M+ seeds of *L. campestre*, about 94% of the wet M+ seeds of *L. campestre*, and 90% of the dry M+ seeds of *L. campestre* remained attached to the mother plant after the wind-induced experiment (Figure 4B). There was no significant difference between *L. draba* and *L. campestre* in relation to wind as a primary dispersal vector (Figure 4B). This study highlights that wind is not the probable primary dispersal vector in both species.

### 2.3. Comparative Mechanical Characterization of Fresh and Degraded Lepidium Fruits: Focus on Stem–Pedicel, Fruit–Pedicel, and Radicle Protrusion Sites

The biomechanical analysis revealed that fresh mature indehiscent fruits of *L. draba* required significantly more force for fruit detachment at the stem–pedicel, fruit–pedicel, and radicle protrusion sites compared to the degraded fruits of *L. draba* (Figure 5). The difference in the force required to detach fruits from the stem–pedicel, fruit–pedicel, and radicle protrusion sites between fresh and degraded fruits of *L. draba* was statistically significant (*p* < 0.0001, Figure 5). When comparing the force applied on fruit opening between the indehiscent *L. draba* fruits, the dehiscent *L. campestre* fruits, and the transgenic *Lepidium campestre* RNAi-LcIND line (artificially generated indehiscent), it was observed that the indehiscent fruits of *L. draba* needed significantly more force (*p* < 0.0001), followed by the transgenic *L. campestre* (*p* < 0.0001, Figure 5D), and the least force was required for the dehiscent fruited *L. campestre* (*p* < 0.0001).

## 3. Materials and Methods

### 3.1. Seed Sources

Seeds for this study were obtained from mature fruits of *Lepidium draba* L. collected from 39.5 N, 26.9 E (Burhaniye, Turkey), and mature seeds of *Lepidium campestre* (L.) W.T. Aiton collected from 56.8 N, 12.9 E (SJömossevägen, Halmstads kommun, Halmstad Municipality, Sweden). Seeds and fruits of these species were collected from plants cultivated in the botanical garden of the University of Osnabrück, Germany. Further, seeds and fruits of the transgenic *Lepidium campestre* line RNAi-LcIND (henceforth Transgenic *Lepidium campestre*), in which RNAi-mediated silencing of the LcINDEHISCENT gene, resulting in the production of indehiscent fruits, were obtained from Friedrich Schiller University Jena, Jena, Germany; for details of cloning, transformation, and plant cultivation procedures, see [21].

### 3.2. Wire-Hook-Induced Force Applied to Detach Seeds and Fruits from the Infructescence

Small-sized wire hooks were placed on the seed and replum junction, and the mass of wire hooks necessary to detach seeds from the replum was used to calculate the force (N) required to detach seeds from the replum using the formula force (N) = mass (in kg) multiplied by acceleration (m/s^−2^); see also Arshad et al. [29]. A total of 200 samples for each type of diaspore (dry mucilaginous seeds, dry M+; wet mucilaginous seeds, wet M+; re-dried mucilaginous seeds, re-dried M+ seeds) from *Lepidium campestre* were tested, providing a robust dataset for analysis. This study aimed to enhance our understanding of the mechanical dynamics influencing diaspore dispersal in *Lepidium campestre*.

### 3.3. Fruit, Fruit Valve, and Seed Resistance to Raindrop Impact

To assess the resistance of fruits, fruit valves, and seeds to raindrop impact, a raindrop test was employed on mature fruits on dry and wet infructescences of both species. A burette with an opening of 1.5 mm in diameter was positioned 50 cm above a single fruit or seed. The number of individual raindrops, with a mean mass of 53.55 ± 0.97 mg, directly impacting the fruit/seed was recorded until fruit abscission (in dehiscent and indehiscent fruits) or seed detachment (in M+ seeds) occurred following the method described by Arshad et al. [29]. M+ represents mucilaginous seeds. Each type of diaspore was subjected to this assessment, with a total of 200 individual samples tested for a comprehensive evaluation of resistance.

### 3.4. Simulated Rainfall Experiment to Assess Seed and Fruit Dispersal

To simulate the effect of rainfall on seed and fruit dispersal, a rain box containing 1 L of water was positioned at a 40 cm height above the infructescence, which contained 50 seeds/fruits. The rain box featured 60 holes with a diameter of 0.4 mm to simulate rainfall. Following the completion of the simulated rainfall, the number of seeds/fruits detached from the replum or pedicle-fruit junctions was counted. Each diaspore type was tested in a total of 200 trials, facilitating a robust dataset for analysis of dispersal mechanisms under simulated rainfall conditions.

### 3.5. Sheep Wool Contact Experiment to Simulate Diaspore Attachment and Detachment from the Infructescence

In the sheep wool contact experiment designed to simulate diaspore attachment and detachment from the infructescence, 200 dry mucilaginous (dry M+), wet mucilaginous (wet M+), re-dried mucilaginous (re-dried M+) seeds, dehiscence (DEH) fruits of *Lepidium campestre*, and indehiscent (IND) fruits of *Lepidium draba* were exposed to four sub-experiments. First, we evaluated diaspore attachment to sheep wool and assessed their adherence ability to sheep fleece, simulating diaspore dispersal by animal fur (exozoochory). The sheep fleece was affixed to a 30 rpm rotating shaker to simulate animal movement, and a maximum of 200 contacts was established. The number of diaspores remaining attached to the sheep fleece was then counted, following the method described by Couvreur et al. [30]. Second, in the same sheep wool contact experiment, we evaluated the detachment ability of diaspores upon contact with sheep fleece. The sheep fleece was affixed to a 30 rpm rotating shaker to simulate animal movement and up to 200 contacts were established. Following the method described by Couvreur et al. [30], we counted the number of diaspores detached upon contact with the sheep fleece. Thirdly, we examined the number of contacts required to detach the diaspores from the replum and finally the frequency of seeds and fruit valves detaching together in dehiscent fruits of *Lepidium campestre* upon contact with sheep wool were evaluated. Dry M+ represents seeds before imbibition, wet M+ represents imbibed seeds, and re-dried M+ represents seeds dried after imbibition.

### 3.6. Wind-Induced Dispersal of Diaspores

To investigate wind-induced dispersal of diaspores, we assessed the detachment of diaspores from fruit infructescences using a fan positioned at a height of 30 cm, generating a wind speed of 4 m/s (see also Arshad et al. [29]). To protect the air flow from external influences, an air chamber with a height of 40 cm was constructed. Infructescences were placed 5 cm in front of the fan for 30 min, and the number of seeds/fruits detached from the infructescences was recorded. Each diaspore type was tested in a total of 200 trials, allowing for a robust analysis of wind-induced dispersal dynamics.

### 3.7. Biomechanical Analysis

The mechanical properties of dispersal units of *Lepidium draba*, *Lepidium campestre*, and transgenic *Lepidium campestre* were determined using a modified Zwick Roell ZwickiLine Z0.5 universal testing machine (ZwickRoell GmbH & Co. KG, Ulm, Germany). All experiments were performed at a speed of 1 mm/min while force and displacement were recorded simultaneously as a method described by Steinbrecher et al. [31] and Hourston et al. [32]. Maximum forces were determined from the force-displacement curves. The force to pull off the pedicel from the stem (n_fresh_ = 45, n_degraded_ = 41), the force to detach the fruit from the pedicel (n_fresh_ = 42, n_degraded_ = 44), and the force to puncture through the pericarp (simulating radicle protrusion) (n_fresh_ = 32, n_degraded_ = 31) were measured for fresh and degraded samples of *L. draba*. A rounded metal probe with a diameter of 0.3 mm was driven into the sample. The force to split open fruits (n_draba_ = 37, n_campestre_ = 32, n_transg_campestre_ = 27) was measured in *L. draba* and *L. campestre* and transgenic *L. campestre*. Samples were clamped on both sides and pulled apart until the rupture occurred [31,32]. To evaluate the impact of after-ripening on biomechanical properties, we collected after-ripened fruits from the mother plants within four months following their maturity. The study specifically focused on fruits that had undergone degradation during this after-ripening period. This approach allowed us to assess any biomechanical changes that may occur as the fruits mature and degrade, providing valuable insights into the mechanical attributes that influence seed dispersal and establishment.

### 3.8. Data Analysis

One-way ANOVA was conducted to analyze the association between treatments for the various dispersal experiments and biomechanical analysis. The data were subjected to one-way ANOVA and Tukey’s honest significant difference test for post hoc comparisons. The rejection threshold for all analyses was set at *p* < 0.05. The results were graphed using R version 4.3.2 (The R Foundation for Statistical Computing) and PRISM v. 7.0a (GraphPad, San Diego, CA, USA).

## 4. Discussion

### 4.1. Adaptive Evolution and Dispersal Strategies in Dehiscent vs. Indehiscent Fruits of Lepidium: Implications for Seed Ecology and Invasion Dynamics

Dehiscent fruits are most likely the ancestral fruit type in Brassicaceae [19,20]. The transition of some plant species from dehiscent to indehiscent fruit could be associated with evolving adaptive traits, particularly in response to germination and dispersal. There are two phases of seed dispersal: primary and secondary. Primary dispersal is defined as the initial dispersal of seeds and fruits from the mother plants [14,33,34]. Any further dispersal after the primary dispersal is referred to as secondary dispersal [14,33,34]. This study highlights the findings of the primary dispersal mechanisms since earlier studies reported that any secondary dispersal mechanism of the study species was associated with seed mucilage of the two species plus fruits of *L. draba* [28]. Our findings highlight differences in the preferred primary dispersal vectors between the indehiscent fruit-producing *L. draba* and the dehiscent fruit-producing *L. campestre*. Our study suggests that indehiscent fruit-producing *L. draba* may less likely rely on primary dispersal vectors. The detachment of fruits from the stem–pedicel and fruit–pedicel may not be facilitated solely by wind, rainfall, and animal attachment in *L. draba*, suggesting that these are less to be likely the primary dispersal vectors for the species. In contrast, the dehiscent fruit-producing *L. campestre* may utilize rainfall or animal contact as primary dispersal vectors, with wind being the least probable dispersal vector in this species. Thus, for the dehiscent fruit-producing *L. campestre*, fruit–replum and fruit–pedicel detachment seems mainly associated with rainfall and animal contact.

Studies on the secondary dispersal mechanisms of the study species, which involve the further dispersal of diaspores after detachment from the mother plant, could provide valuable insights into the broader dispersal dynamics of these species. In the case of *L. campestre*, the mature fruit opens up and releases two seeds that become mucilaginous and sticky upon wetting [23]; Figure 1. Conversely, the closely related *L. draba* develops non-fleshy, indehiscent fruits [15,22]; Figure 1. In this scenario, the fruit valves remain, enclosing the two seeds at fruit dispersal (Figure 1B), and the seeds germinate after the pericarp-mediated chemical dormancy is released [24], with the whole fruit serving as the dispersal unit [24,28,35,36]; Figure 1B. While indehiscent fruits in Brassicaceae typically contain non-sticky seeds, *L. draba* exhibits sticky seeds similar to *L. campestre*, indicating that the shift from dehiscent to indehiscent fruit may have occurred relatively recently in the evolution of the genus *Lepidium* [20]. The production of mucilaginous and sticky seeds can offer advantages for the plant’s survival, either by controlling the timing of germination or by enabling the seeds to disperse farther [28,37]. The secondary seed dispersal associated with mucilage revealed that differences in pectin and cellulose contents in the outer seed coat layer of the mucilaginous seeds of *L. campestre* and *L. draba* have led to significant variations in water uptake and retention between the two species [28,38]. These findings show that the mucilage increased water uptake in both species, with *L. campestre* seeds absorbing a remarkable 836% more water compared to 75% in *L. draba*. Surprisingly, despite this substantial difference, the presence of mucilage had a minimal impact on seed germination for both species. The results indicate that re-dried mucilaginous seeds exhibit a significantly stronger adherence to the ground compared to both wet mucilaginous seeds and other diaspore types examined in the study species [28]. This enhanced anchorage can be attributed to the unique properties of the mucilage, which, when re-dried, demonstrates superior adherence capabilities [28]. Such strong adherence is crucial for seed retention, since these seeds are less susceptible to displacement from environmental factors such as rainfall, wind, or animal interactions. In contrast, the wet mucilaginous seeds and other diaspore types of the study species tend to be more easily removed under similar conditions [28]. These findings highlight the ecological advantage provided by the re-dried mucilaginous seeds, reinforcing their potential for successful establishment in competitive environments.

Moreover, the study revealed interesting insights into the dispersal strategies of these two invasive weed species. While the seeds and fruits of *L. draba* demonstrated a higher potential for dispersal via water and wind as a secondary dispersal vectors, the seeds of *L. campestre* exhibited a tendency to adhere to animals. Furthermore, *L. campestre* displayed significantly stronger adherence to sand particles (1872%) compared to *L. draba* seeds (445%), indicating a greater likelihood of local secondary dispersal and potential adhesion to animals or soil particles. Mohammed and Mummenhoff [28] highlight the evolution of mucilage production as a facilitator of diverse dispersal methods in these invasive weed species. This may offer a possible explanation for the observed differences in their distribution patterns and could have broader implications for understanding the ecological dynamics of invasive plants. These findings shed light on the adaptive capabilities of these species and provide valuable insights into the mechanisms underlying their successful establishment and spread in various ecosystems.

### 4.2. Disparities in Primary Seed Dispersal Mechanisms of Lepidium campestre and Lepidium draba: The Role of Animal Movement, Rainfall, and Wind

In this study, we have observed distinct patterns of primary seed dispersal in *L. campestre* and *L. draba*. Our findings suggest that *L. campestre* seeds are more likely to be dispersed or detached from the mother plant through animal movement and rainfall, with wind being the least probable dispersal vector for this species (Figure 3B and Figure 4). This implies that when animals move through the fields where *L. campestre* is present, there is a high potential for the seeds to be deposited on the ground, facilitated by animals acting as primary dispersal vectors (Figure 3B). Additionally, the occurrence of rainfall greatly increases the likelihood of seeds falling on the ground (Figure 4A). Our study highlights these two mechanisms as the primary dispersal vectors for *L. campestre*. The study further emphasizes that during primary dispersal, approximately 50–60% of the seeds and fruit valves detach together from the replum (Figure 2A). This suggests a high probability of seeds and fruit valves detaching together during primary dispersal events, such as rainfall and animal interactions. Moreover, it was observed that wet dehiscent fruits (about 60%) exhibited a higher tendency for seeds and fruit valves detaching together compared to dry dehiscent fruits (50%) in *L. campestre* (Figure 2A). This indicates that the rainy season and associated animal movements could significantly increase the likelihood of seeds and fruit valves detaching together in this species. Furthermore, the research highlights that in cases where seeds remain attached to the replum on the mother plant, dry seeds (before the rainy season) and re-dried seeds (re-drying after the rainy season) demonstrate a stronger attachment to the replum compared to the wet mucilaginous seeds (during rainfall) of this species (Figure 2B). This further supports the potential for rainfall to facilitate the detachment of diaspores from the mother plant in *L. campestre*.

On the other hand, the closely related *L. draba*, which produces indehiscent fruits, is less likely to be primarily dispersed through wind, rainfall, but rarely relies on animal contacts (Figure 3B and Figure 4A,B). Our study indicates that animal contact in areas where mature *L. draba* plants grow may have some effects on fruit detachment from the mother plant (Figure 3B). However, rainfall alone does not effectively detach fruits from the mother plant to act as a primary dispersal vector in this species (Figure 4A). Also, in our observations, we noted that specific wind speeds did not lead to fruit detachment from the mother plant (Figure 4A). This finding suggests that *L. draba* may predominantly depend on animal movements for primary dispersal (Figure 3B). Considering that *L. draba* is primarily found in disturbed habitats [39], it is probable that animal movement may result in the breaking of the infructescence, causing the entire infructescence to fall to the ground and act as a primary dispersal vector.

### 4.3. Effects of After-Ripening on Biomechanical Analysis and Dispersal Mechanisms in Lepidium draba: Insights into Fruit Degradation and Germination Dynamics

The biomechanical analysis conducted on freshly harvested and degraded fruits of *L. draba* reports fascinating findings (Figure 5A–D). Our experiments using a puncture force device revealed that degraded fruits required significantly less force for detachment from the mother plants at key points such as the fruit–pedicel junction, stem–pedicel junction, and radicle protrusion sites, as well as for opening the fruit (Figure 5A–D). Thus, freshly mature fruits of *L. draba* may remain attached to the mother plants until they undergo a period of after-ripening (post-maturation) during which fruit coat (pericarp) degradation occurs. This occurs during the summer season and requires approximately four months (personal observation). As a result, for primary dispersal, the freshly mature fruits of *L. draba* may undergo a form of after-ripening (aging and degradation), which is beneficial for both dispersal and germination. Thus, any primary dispersal vector would need to wait until after-ripening or post-maturation stage has been reached. Our observations indicate that following post-maturation, the entire infructescence of the plant falls to the ground when subjected to animal movements. Subsequently, under favorable environmental conditions, the seeds have the potential to germinate out of the fruit once fallen to the ground. Additionally, it is important to consider the potential involvement of secondary dispersal vectors, as outlined by Mohammed and Mummenhoff [28], which may contribute to secondary dispersal opportunities when the entire infructescence falls to the ground.

### 4.4. Differential Dormancy Mechanisms and Timing of Germination in Lepidium: Implications for Seed Management and Ecological Strategies

Previous study demonstrates that the seeds of *L. draba* germinate directly out of the fruit, and there is a specific timing for seed germination. Specifically, the fresh fruits of *L. draba* exhibited a higher mechanical resistance, indicating that the fresh mature fruits may not germinate and may employ a dormancy strategy governed by chemical-induced fruit coat dormancy [24]. A previous study sheds light on the presence of chemical inhibitors, particularly abscisic acid (ABA), in the fruit coat of *L. draba*, leading to pericarp-mediated chemical dormancy. The research demonstrates that washing *L. draba* fruits and after-ripening (post maturation) can remove these inhibitors, significantly enhancing germination rates, and highlights the physiological roles of ABA and gibberellins in the dormancy and germination processes of these invasive species, offering insights that could inform weed management strategies in agricultural contexts. In the case of *L. campestre*, the freshly harvested seeds exhibit physiological dormancy that released through after-ripening [25].

### 4.5. Dispersal Mechanisms and Species Distribution: Impacts of Primary and Secondary Dispersal Vectors on the Invasive Potential of Lepidium draba and Lepidium campestre

It is important to consider the native and introduced distribution areas of the species in comparison to the dispersal mechanisms identified in this study, as dispersal directly influences species distribution. *L. draba*, native to central Asia and Siberia, including the Balkan Peninsula, Georgia, Armenia, Azerbaijan, Turkmenistan, Kazakhstan, southern Russia, Turkey, Israel, Syria, Iraq, and Iran, experiences a Mediterranean and continental-influenced climate, featuring warm summers but very cold and dry winters [39,40,41,42,43]. In contrast, *L. campestre*, native to Europe, is commonly found in the humid/humid nemoral zone with reliable and sufficient rainfall supply in countries across the Nordic region [18,26,27,44]. *L. draba* is most prevalent in arable land in Europe [39,45] and has become established as an agricultural weed in Canada, Australia, and the United States [39,46,47,48,49]. In the USA, Canada, and Australia, *L. draba* is classified as noxious and invasive weeds, posing significant challenges in terms of control and eradication [16,17,39]. In contrast, the distribution patterns of *L. campestre* suggest its invasive nature in regions with sufficient rainfall [26]. The distribution of the study species is strongly influenced by both primary and secondary dispersal vectors. Primary dispersal vectors, such as animals and rainfall, provide a significant advantage for *L. campestre* compared to *L. draba* in terms of the species distribution. Consequently, *L. campestre* primarily relies on primary dispersal to enhance its invasiveness and expand its distribution range. On the other hand, the highly invasive and noxious weed, *L. draba*, predominantly depends on secondary dispersal mechanisms, such as seed and fruit dispersal by wind and rainfall. This study reports the greater importance of secondary dispersal over primary dispersal in *L. draba*, possibly due to the fruit and seed morphology of this species being more conducive to secondary dispersal than primary dispersal (Figure 1). As a result, once mature and undergoing post-maturation, the phenomenon of the entire infructescence falling to the ground provides an opportunity for secondary dispersal, which is key for this species. The study emphasizes that examining the distribution range of invasive species should consider the interplay of both primary and secondary dispersal mechanisms, which can vary from species to species. Other findings suggest that secondary dispersal can lead to longer-distance dispersal compared to primary dispersal [9,50,51,52,53], and it is likely that the widespread distribution of *L. draba* can be attributed to the high likelihood of secondary dispersal. One of the reasons could be that the strong adhesive properties of mucilage play a crucial role in facilitating the long-distance dispersal of seeds via animal fur. When mucilage adheres to fur, its re-drying further enhances this attachment, allowing the seeds to remain securely attached during the animal’s movements. This mechanism not only aids in the dispersal of the species but also increases the likelihood of seed germination in new locations, thereby promoting colonization. The potential for such long-range dispersal highlights the evolutionary advantage conferred by the mucilaginous coating, suggesting its significance in the ecological success of the species in various environments [28]. Consequently, while primary dispersal seems more crucial for *L. campestre*, secondary dispersal confers significant benefits for *L. draba* over longer distances.

## 5. Conclusions

The distribution range of invasive species is significantly correlated with their dispersal mechanisms. Our study suggests that while *Lepidium campestre* mainly relies on animal and rainfall for primary dispersal, *L. draba* fruits exhibit minimal dependence on animal movement as primary dispersal vectors. Instead, the dispersal of *L. draba* fruits predominantly occurs after they fall to the ground (post-maturation or due to animal contacts) and are further dispersed by secondary vectors, such as wind and rainfall. Therefore, secondary dispersal vectors play a more significant role than primary dispersal vectors for *L. draba*, providing insights into the extensive distribution of the species. The invasiveness of *L. draba* seems primarily facilitated by its reliance on secondary dispersal of its fallen infructescences, allowing for dispersal over larger distances compared to primary dispersal. Our study highlights the divergent dispersal strategies of the closely related dehiscent fruit-producing *L. campestre* and the indehiscent fruit-producing *L. draba*, both of which contribute to the invasive success of these species within their distribution range. Consequently, proactive measures should be considered to control the invasiveness of the study species, taking into account their specific dispersal mechanisms.

## Figures and Tables

**Figure 1 plants-14-00446-f001:**
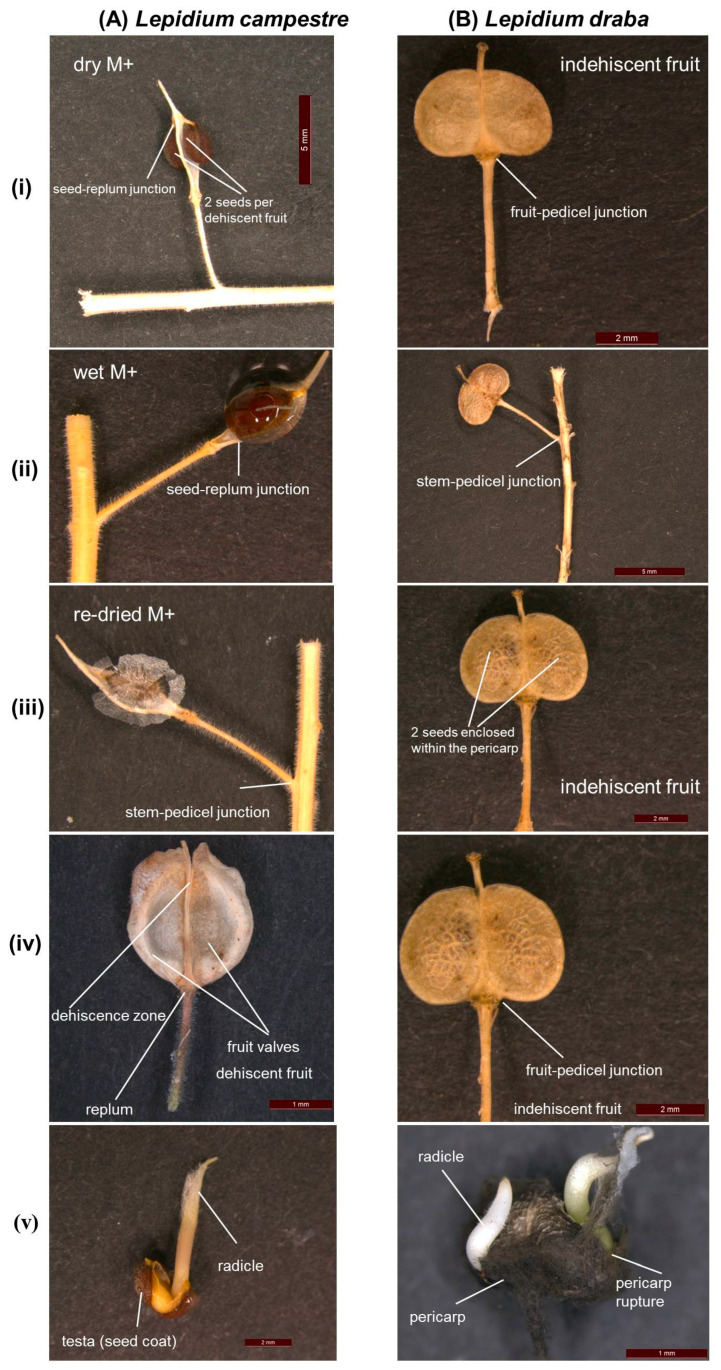
Detailed photographical illustration of true dispersal units and contrasting germination processes in species under study: (**A**) seeds in dehiscent fruit *L. campestre* and (**B**) indehiscent fruits of *L. draba*. (i) Depiction of dry mucilaginous (M+) seeds of *L. campestre* and indehiscent fruit of *L. draba*. M+ seeds getting mucilaginous upon wetting. (ii) Illustration of wet mucilaginous (M+) seeds of *L. campestre* and indehiscent fruits of L. draba. (iii) Re-dried mucilaginous (M+) seeds of *L. campestre* and indehiscent fruit of *L. draba* showing 2 seeds enclosed within the pericarp. (iv) Comparison of dehiscent (*L. campestre*) and indehiscent (*L. draba*) fruit form. (v) Contrasting seed germination processes, with *L. campestre* characterized by seeds as true dispersal and germinating units, while in *L. draba* indehiscent fruits are the dispersal units and the enclosed seeds are germinating directly out of the fruit.

**Figure 2 plants-14-00446-f002:**
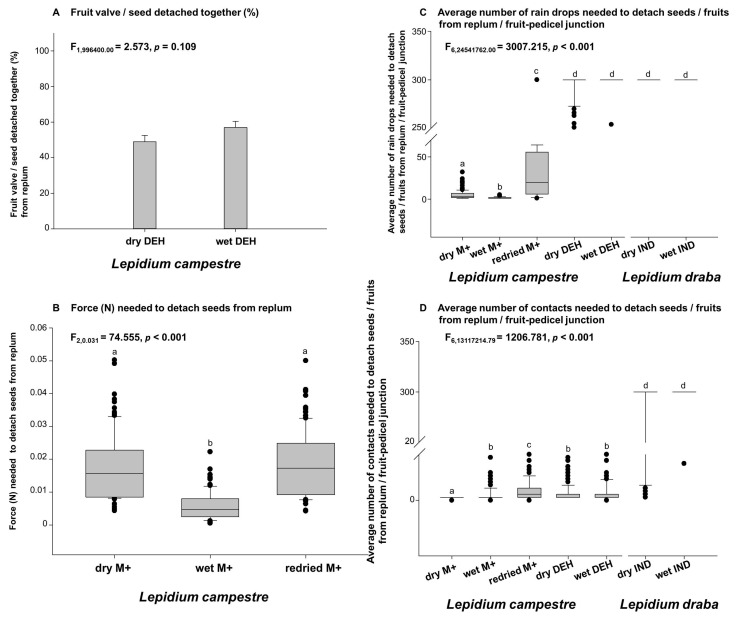
Seed and fruit detachment frequency, the force required to detach seeds from replum in dehiscent (DEH) *L. campestre* (*Lc*) fruits (**A**,**B**), and the results of dispersal experiments using raindrop simulation and mechanical contact to detach *L. campestre* seeds from the replum and *L. draba* indehiscent (IND) fruits from the pedicel. (**A**) The percentage of fruit valve dehiscent and seed abscission occurring simultaneously. The seeds and fruits were in contact with sheep wool fixed on a rotatory shaker with a contact speed of 30 rpm and a maximum of 200 contacts were applied. (**B**) The force needed to detach seeds from the replum of *L. campestre* DEH fruits. Small wire hooks were placed on the seed and replum junction, and the mass of the wire hooks needed to detach seeds from the replum was used to calculate the force (N) required for detachment. (**C**) The impact of raindrops on seed and fruit detachment. The number of artificial raindrops required to detach *L. campestre* seeds from the replum and *L. draba* fruits from the pedicel is given. Water drops with a diameter of 1.55 mm were applied to the seeds and fruits from a height of 40 cm. A maximum of 300 raindrops (75 drops per minute) was applied, and the number of raindrops needed to detach seeds and fruits was recorded. (**D**) The impact of animal contacts on seed and fruit detachment. The average number of artificial contacts needed to detach *L. campestre* seeds from the replum and *L. draba* fruits from the pedicel. Seeds and fruits were in contact with sheep wool fixed on a rotatory shaker with a contact speed of 30 rpm, and the number of contacts needed for detachment was recorded. For further details, refer to the Materials and Methods section. Abbreviations: DEH—dehiscent fruits; M+—mucilaginous seeds; IND—indehiscent fruits. *n* = 200 for each treatment. Different letters (a, b, c, d) indicate significantly different means as determined by the Tukey pairwise multiple comparison test (*p* < 0.05). See the M & M chapter for detailed description of dispersal experiments.

**Figure 3 plants-14-00446-f003:**
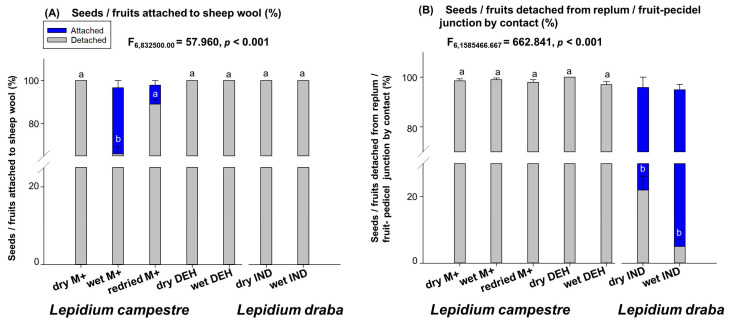
Mechanical experiments to mimic seed and fruit attachment to and detachment by animals (sheep wool). (**A**) Illustration of diaspores’ attachment to animal fur (sheep). Seeds and fruits were in contact with sheep wool attached to a rotatory shaker (30 rpm), and the number of seeds/fruits attached to sheep was counted with a maximum of 200 contacts. (**B**) Analysis of diaspore detachment from fruit infructescence by animals. Seeds and fruits were in contact with sheep wool fixed on a rotatory shaker with a contact speed of 30 rpm, and the number of contacts needed to detach seeds and fruits was recorded. Abbreviations: dry DEH—dry dehiscent fruits; wet DEH—wet dehiscent fruits; dry M+—dry mucilaginous seeds; wet M+—wet mucilaginous seeds; re-dried M+—re-dried mucilaginous seeds; dry IND—dry indehiscent fruits; wet IND—wet indehiscent fruits. *n* = 200 for each treatment. Different letters (a, b) indicate significantly different means as determined by the Tukey pairwise multiple comparison test (*p* < 0.05). For further details, refer to the Materials and Methods section.

**Figure 4 plants-14-00446-f004:**
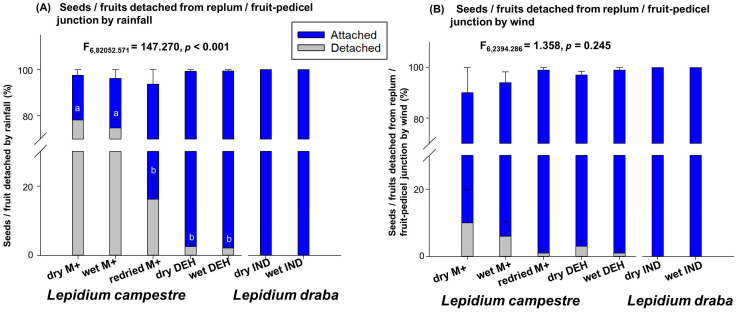
Water drop and fan experiments to mimic seed and fruit detachment of seeds and fruits from replum/fruit–pedicel junction by rainfall (**A**) and wind (**B**). (**A**) Analysis of diaspores’ detachment from fruit infructescence by rainfall. A rain box containing 1 L of water was placed 40 cm above the infructescence with 50 seeds/fruits (Arshad et al. [29]). The rain box had 60 holes with a 0.4 mm diameter to simulate rainfall, and after the rain fell, the number of seeds/fruits detached from the replum or pedicle–fruit junctions was counted. (**B**) Analysis of diaspores’ detachment from fruit infructescence by wind (Arshad et al. [29]). A fan with a height of 30 cm was used to detach seeds/fruits from infructescence with a wind speed of 4 m/s. The air flow from the fan was protected from any external air influence by building an air chamber with a height of 40 cm. Infructescences were placed with a distance of 5 cm in front of a fan for 30 min, and the number of seeds/fruits detached from the infructescences was documented. Abbreviations: dry DEH—dry dehiscent fruits; wet DEH—wet dehiscent fruits; dry M+—dry mucilaginous seeds; wet M+—wet mucilaginous seeds; re-dried M+—re-dried mucilaginous seeds; dry IND—dry indehiscent fruits; wet IND—wet indehiscent fruits. *n* = 200 for each treatment. Different letters (a, b) indicate significantly different means as determined by the Tukey pairwise multiple comparison test (*p* < 0.05). For further details, refer to the Materials and Methods section.

**Figure 5 plants-14-00446-f005:**
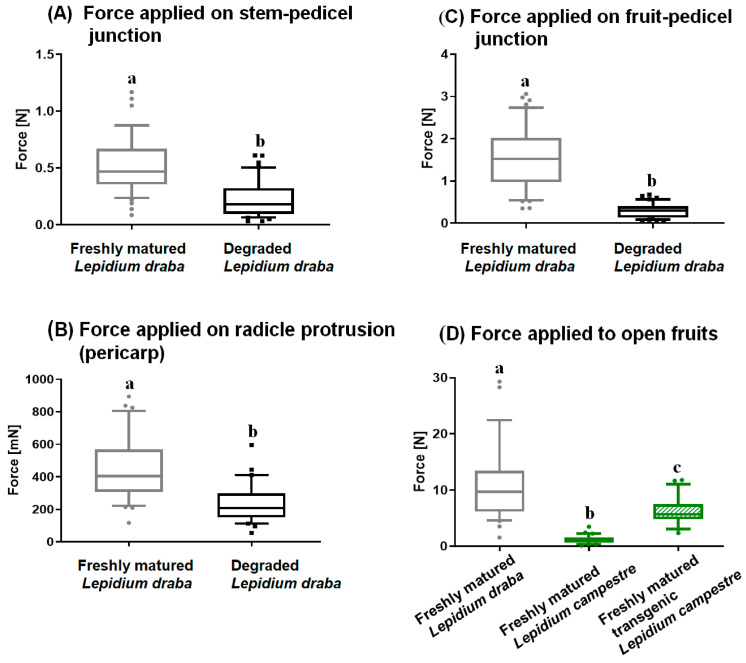
The biomechanical analysis applied to detach seeds/fruits from the mother plant. (**A**) Force applied on the stem–pedicel junction on both freshly harvested and degraded fruits of *L. draba*. (**B**) Force applied on the radicle protrusion site of the pericarp on both freshly harvested and degraded fruits of *L. draba*. (**C**) Force applied on the fruit–pedicel junction on both freshly harvested and degraded fruits of *L. draba*. (**D**) Measurement of the force applied to open fruits of freshly harvested fruits of *L. draba*, freshly harvested fruits of *L. campestre*, and freshly harvested fruits of transgenic *L. campestre* (artificially indehiscent fruits). The green colour represents *L. campestre* in both its wild type and artificially induced indehiscent fruits.The transgenic *L. campestre* seeds were obtained from Lenser and Theissen [21]. Freshly mature fruits refer to newly maturing fruits, whereas degraded fruits refer to fruits that remained on the mother plant for four months after reaching maturity. Different letters (a–c) indicate significantly different means as determined by the Tukey pairwise multiple comparison test (*p* < 0.05). For further details, refer to the Materials and Methods section.

## Data Availability

The original contributions presented in this study are included in the article. Further inquiries can be directed to the corresponding author.

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
