# Peer review of "Differential Primary Seed and Fruit Dispersal Mechanisms and Dispersal Biomechanics in Invasive Dehiscent and Indehiscent-Fruited Lepidium Species"

_plants, 2025, doi:10.3390/plants14030446_

Round 1
Reviewer 1 Report
Comments and Suggestions for Authors
The title is clear and informative but could be more concise.
The abstract is comprehensive but could be more concise. I recommend shortening and focusing on key findings and implications.
The introduction is well-designed, providing a solid foundation for understanding the importance of dispersal in the life cycle of plants. Authors should consider relocating Figures 1 and 2 to the materials or discussion sections.
Further, clarify the discussion on the ecological and evolutionary implications of the findings.
Comments on the Quality of English Language
Check grammar and writing style for a more fluent and error-free text.
Author Response
Detailed response to the Reviewer
Reviewer #1
Comments and Suggestions for Authors
- The title is clear and informative but could be more concise.
The authors appreciate the feedback regarding the manuscript title, we have changed it to be more concise to "Differential Primary Seed and Fruit Dispersal Mechanisms and Dispersal Biomechanics in Invasive Dehiscent and Indehiscent-Fruited Lepidium Species."
- The abstract is comprehensive but could be more concise. I recommend shortening and focusing on key findings and implications.
We have made the abstract more concise as suggested. It is now also shorter, consisting of 192 words and efficiently captures the key findings of our study while maintaining clarity and conciseness.
- The introduction is well-designed, providing a solid foundation for understanding the importance of dispersal in the life cycle of plants. Authors should consider relocating Figures 1 and 2 to the materials or discussion sections.
We would like to retain these figures in the introduction section as provide insights into the morphology of the study plants early on is essential for helping readers to understand the context of our research. A thorough introduction to the plants' morphology, as well as their seed and fruit architecture, sets the foundation for a clearer comprehension of the subsequent details in the manuscript. Therefore, we opted to present this information at the beginning to enhance the overall coherence of the study.
- Further, clarify the discussion on the ecological and evolutionary implications of the findings.
In the first subtopic of the discussion section, titled "4.1. Adaptive Evolution and Dispersal Strategies in Dehiscent vs. Indehiscent Fruits of Lepidium: Implications for Seed Ecology and Invasion Dynamics," we provide a detailed examination of these implications. For instance, lines 373-376 highlight that dehiscent fruits are likely the ancestral fruit type in Brassicaceae, and we discuss how the transition to indehiscent fruit types in certain species may be linked to adaptive traits that enhance germination and dispersal success. Furthermore, between lines 397-412, we present a comparative analysis of L. campestre and L. draba, illustrating the common fruit characteristics such as mucilaginous and sticky seeds provide important insights into their evolutionary trajectories. This comparison suggests that the evolution from dehiscent to indehiscent fruits may have occurred relatively recently within the genus Lepidium, which could have significant implications for their dispersal strategies. The discussion also connects the differences in distribution patterns of the studied species to their preferred dispersal vectors, reaffirming our commitment to exploring both ecological and evolutionary contexts. We appreciate your feedback, as it has prompted us to ensure that these critical points are articulated more clearly in the manuscript.
- Comments on the Quality of English Language
We went through the manuscript and have further improved the grammar, coherence, and overall fluency of the manuscript contribute to its readability and clarity. We appreciate your consideration and are open to any specific suggestions you may have for further improvement.
- Check grammar and writing style for a more fluent and error-free text.
We went through the manuscript and have further improved the grammar, coherence, and overall fluency of the manuscript significantly enhance its readability and clarity. We truly appreciate your consideration and welcome any specific suggestions you may have for further improvement.
- Does the introduction provide sufficient background and include all relevant references? Can be improve.
The introduction section has been further improved by providing additional background information. Kindly find the revised version of the manuscript. For instance, we have included the significance of the INDEHICENT (IND) gene, which is crucial for the formation of the separation and lignified layer that regulates fruit dehiscence (Mummenhoff et al., 2009; Mühlhausen et al., 2013). We also discuss how down regulating the expression of this gene in the transgenic L. campestre RNAi-LcIND line leads to the production of indehiscent fruits (Lenser and Theissen, 2013). Furthermore, we have incorporated other important references, such as Willis et al. (2014), to strengthen the background provided in the introduction.
Reviewer 2 Report
Comments and Suggestions for Authors
This is one of a series of investigations on the dispersal of two weedy Lepidium species with contrasting dispersal strategies. The authors have sought to determine the relative importance of primary dispersal, where diaspores are removed from the plant itself, to secondary dispersal, where diaspores are picked up by other agents and moved away from the vicinity of the parent plant. One species has dehiscent fruits and releases or exposes its seeds to potential dispersal agents; the other has indehiscent fruits that are rarely removed from the plant by dispersal agents, but after they mature, collapse, and fall from the plant, they may be picked up by wind, water, and other agents, and seeds can germinate within the fruits.
The experiments and observations are all pertinent to their questions, and well-described, especially in the legends of the figures. The statistics used all seem appropriate, and it is good they show all data even those which do not show significant differences. There are numerous self-citations but all are warranted as they are about different aspects of this same system.
I think the authors should offer an explanation of why the re-dried mucilaginous seeds are less likely to detach – it seems likely to me that it is because they became glued to the infructescence stalk. I do not think they looked at how likely seeds were to detach from animal fur, which would also have been interesting, as wet seeds picked up thanks to their sticky mucilage might dry on the fur, leading to them traveling farther than if they are loosely attached.
My one objections is that the authors assume that secondary dispersal in the one species may lead to longer distance dispersal than its primary modes and (it seems to be implied) dispersal in the other species. One of the authors is known for their work on animal-assisted dispersal, which can often result in diaspores moving long distances from parent plants. I would like to see more discussion of why they think this is the case, as the mucilaginous seeds of the other species can lead the seeds to adhere to animals that may take those seeds large distances.

I recognize that the authors may have English as an additional language, and so I point out a number of miswordings throughout the pdf. I have highlighted them on the pdf and point them out line by line. Some things are consistent misuses throughout (like using indehiscence fruit rather than indehiscent), so I have not corrected them more than once or twice, and recommend they be changed throughout. Attention needed to lit cited – esp. odd spacing and lack of italics when needed.
Line 15 – should be “dehiscent” and “indehiscent” as these are the adjective forms – this change should be made throughout the manuscript
Line 17 – delete the word “into”
Line 19 – replace “a” with “is rarely involved in”
Line 23 – secondary dispersal by what? Needed to make the sentence make sense. Perhaps say “secondary dispersal via mucilage on seeds”
Line 178 – data is a plural, so use “were” not “was”
Line 186 – detaching
Line 230 – seed not seeds
Line 239 – use “adherence to animals” rather than “ability”
Line 360 – radicle misspelled
Line 377 – two phases of seed dispersal: primary and secondary.
Line 380 – Eliminate word “However”
Line 382 – As earlier studies reported that
Line 389 – comma after “rainfall”
Line 443 – seeds falling on the ground
Line 480 – Suggestion: put a period after (post maturation) and eliminate the rest of the sentence. It is repetitive of the next sentence.
Line 490 – once fallen to the ground
Line 510 – replace “and it” with “that”
Line 512 – de-italicize the latin species names
Line 547 – longer-distance dispersal
Line 563 – dispersal of its fallen infructescences,
Author Response
Detailed response to the Reviewer
Reviewer #2
Comments and Suggestions for Authors
This is one of a series of investigations on the dispersal of two weedy Lepidium species with contrasting dispersal strategies. The authors have sought to determine the relative importance of primary dispersal, where diaspores are removed from the plant itself, to secondary dispersal, where diaspores are picked up by other agents and moved away from the vicinity of the parent plant. One species has dehiscent fruits and releases or exposes its seeds to potential dispersal agents; the other has indehiscent fruits that are rarely removed from the plant by dispersal agents, but after they mature, collapse, and fall from the plant, they may be picked up by wind, water, and other agents, and seeds can germinate within the fruits.
The experiments and observations are all pertinent to their questions, and well-described, especially in the legends of the figures. The statistics used all seem appropriate, and it is good they show all data even those which do not show significant differences. There are numerous self-citations but all are warranted as they are about different aspects of this same system.
- I think the authors should offer an explanation of why the re-dried mucilaginous seeds are less likely to detach – it seems likely to me that it is because they became glued to the infructescence stalk. I do not think they looked at how likely seeds were to detach from animal fur, which would also have been interesting, as wet seeds picked up thanks to their sticky mucilage might dry on the fur, leading to them traveling farther than if they are loosely attached.
We have incorporated your suggestions and added the following paragraph to improve the manuscript: The results indicate that re-dried mucilaginous seeds exhibit a significantly stronger adherence to the ground compared to both wet mucilaginous seeds and other diaspore types examined in the study species [28]. This enhanced anchorage can be attributed to the unique properties of the mucilage, which, when re-dried, demonstrates superior adherence capabilities [28]. Such strong adherence is crucial for seed retention, as these seeds are less susceptible to displacement from environmental factors such as rainfall, wind, or animal interactions. In contrast, the wet mucilaginous seeds and other diaspora types of the study species, tend to be more easily removed under similar conditions [28]. These findings highlight the ecological advantage provided by the re-dried mucilaginous seeds, reinforcing their potential for successful establishment in competitive environments. Further, we encourage you to refer to Figures 3 and 4 in the manuscript for a detailed analysis of our findings on detach and attach to animal fur.
- My one objections is that the authors assume that secondary dispersal in the one species may lead to longer distance dispersal than its primary modes and (it seems to be implied) dispersal in the other species. One of the authors is known for their work on animal-assisted dispersal, which can often result in diaspores moving long distances from parent plants. I would like to see more discussion of why they think this is the case, as the mucilaginous seeds of the other species can lead the seeds to adhere to animals that may take those seeds large distances.
We have incorporated your suggestions and added the following paragraph to improve the manuscript: The strong adhesive properties of mucilage play a crucial role in facilitating the long-distance dispersal of seeds via animal fur. When mucilage adheres to fur, its re-drying further enhances this attachment, allowing the seeds to remain securely attached during the animal's movements. This mechanism not only aids in the dispersal of the species but also increases the likelihood of seed germination in new locations, thereby promoting colonization. The potential for such long-range dispersal highlights the evolutionary advantage conferred by the mucilaginous coating, suggesting its significance in the ecological success of the species in various environments [28].
- I recognize that the authors may have English as an additional language, and so I point out a number of miswordings throughout the pdf. I have highlighted them on the pdf and point them out line by line. Some things are consistent misuses throughout (like using indehiscence fruit rather than indehiscent), so I have not corrected them more than once or twice, and recommend they be changed throughout. Attention needed to lit cited – esp. odd spacing and lack of italics when needed.
We have addressed all instances of miswording throughout the manuscript, including the terms "dehiscence" and "indehiscence" as you highlighted. We went through the manuscript and have further improved the grammar, coherence, and overall fluency of the manuscript contribute to its readability and clarity. We appreciate your consideration and are open to any specific suggestions you may have for further improvement.
- Line 15 – should be “dehiscent” and “indehiscent” as these are the adjective forms – this change should be made throughout the manuscript
The correction has been made as requested throughout the ms
- Line 17 – delete the word “into”
The correction has been made as requested.
- Line 19 – replace “a” with “is rarely involved in”
The correction has been made as requested.
- Line 23 – secondary dispersal by what? Needed to make the sentence make sense. Perhaps say “secondary dispersal via mucilage on seeds”
The correction has been made as requested.
- Line 178 – data is a plural, so use “were” not “was”
The correction has been made as requested.
- Line 186 – detaching
The correction has been made as requested.
- Line 230 – seed not seeds
The correction has been made as requested.
- Line 239 – use “adherence to animals” rather than “ability”
The correction has been made as requested.
- Line 360 – radicle misspelled
The correction has been made as requested.
- Line 377 – two phases of seed dispersal: primary and secondary.
The correction has been made as requested.
- Line 380 – Eliminate word “However”
The correction has been made as requested.
- Line 382 – As earlier studies reported that
The correction has been made as requested.
- Line 389 – comma after “rainfall”
The correction has been made as requested.
- Line 443 – seeds falling on the ground
The correction has been made as requested.
- Line 480 – Suggestion: put a period after (post maturation) and eliminate the rest of the sentence. It is repetitive of the next sentence.
The reviewer's comments have been paraphrased and compiled to enhance clarity and coherence. Two reviewers' insights have been integrated, resulting in a more polished and comprehensive set of feedback.
- Line 490 – once fallen to the ground
The correction has been made as requested.
- Line 510 – replace “and it” with “that”
The correction has been made as requested.
- Line 512 – de-italicize the latin species names
The correction has been made as requested.
- Line 547 – longer-distance dispersal
The correction has been made as requested.
- Line 563 – dispersal of its fallen infructescences,
The correction has been made as requested.
Reviewer 3 Report
Comments and Suggestions for Authors
Line 15. Is Lepidium draba an invasive species, or is L. campestre the only species that is invasive? Not clear. Line 25 seems to be saying that L. draba also is invasive.
Line 30. Not very clear. Change to “ Process in the popylation dynamics of plant species, shaping…”
Line 32. Not clear. Change to “dispersal is that it facilitates colonization …”
Line 47. In the Discussion, I learned that both species of invasive. Change to “dispersal of the two invasive closely-related species…” I suggest that you move lines 513-530 to the Introduction, which will give the reader a better idea of why you are interested in dispersal of these two species.
Line 86. Change “of the two species “ to “L. campestre” Your figure 2 shows mucilage only for L. campestre.
Line 103. Change ”(L.)” to “L.”
Line 105. Insert “were” before “collected”
Line 106. Delete “propagated and”
Line 108. Not clear what you did with the transgenic Lepidium campestre seeds. Did you do experiments with them?
Line 113. Delete “method”
Line 114. After “infructescences” should you insert “of both species” Of, did you only do the raindrop test with one of the species? Not clear
Line 137. Change “were” to “was” maximum was
Line s156-157. Delete the last sentence of the paragraph. This is something you might say in the Discussion.
Line 164. Need to say what the different types of diaspore of L. campestre are.
Line 172. Insert a comma before “while”
Line 175. In this section of the methods you do not tell the reader how you studies after-ripening of the seeds. However, you talk about after-ripening in the Results. Need to provide the methods for the after-ripening studies.
Line 184. I found it very difficult to follow and understand your Results. It seems that you have not presented the Results in the same order that you present the Methods. You need to either put the methods or the results in a different order, that is they need to be in the same order.
Lines 186-196. Since you talk about raindrops in the first section of the methods, you should be giving the results from the raindrop test here.
Line 187. What force are you talking about? Raindrops falling? Not clear – heard to follow/understand.
Line 187. Delete “was observed”
Line 190. Delete “the results indicated that”
Line 195. What force? Is this section about the results of puncture test?
Lines 223-230. Now, I find the results from the raindrop test. I strongly recommend that you arrange the topics in the methods and in the results in the same order. A reader should be able to read the first section of the methods and then turn to the first section of the results and learn what happened during the first experiment. Now, your results are a big mixture of topics that is not in the same order as the topics in the methods. Please make the reading of your interesting manuscript an easy experience for the reader.
Line 232. Delete “it was found that”
Line 311. Delete “the study reported that”
Line 336. Delete “The results showed that”
Line 370. Near the first of the Discussion, you need to tell the reader if your hypothesis was support (or not) and talk about the data that support (or not) your hypothesis.
Lines 474-475. Your need to provide the methods for after-ripening.
Line 554, change “influenced” to “correlated”
Author Response
Detailed response to the Reviewer
Reviewer #3
Comments and Suggestions for Authors
- Line 15. Is Lepidium draba an invasive species, or is campestre the only species that is invasive? Not clear. Line 25 seems to be saying that L. draba also is invasive.
We have revised line 15 as follows: “……the two invasive dehiscent fruit producing Lepidium campestre and the indehiscent fruit producing Lepidium draba was conducted”. Please take a moment to review the revised manuscript. Thank you!
- Line 30. Not very clear. Change to “Process in the popylation dynamics of plant species, shaping…”
We have corrected the text as suggested. Additionally, in response to the reviewers' comments, we have revised the phrase "Dispersal is a vital process..." to "Dispersal of seeds or fruits is a vital process...". We believe this change provides greater clarity and specificity.
- Line 32. Not clear. Change to “dispersal is that it facilitates colonization …”
We appreciate the reviewer’s comments and have made the necessary corrections. This helped us to enhance the clarity and quality of our manuscript.
- Line 47. In the Discussion, I learned that both species of invasive. Change to “dispersal of the two invasive closely-related species…” I suggest that you move lines 513-530 to the Introduction, which will give the reader a better idea of why you are interested in dispersal of these two species.
We have corrected the mention of the dispersal of the two invasive, closely related species. In relation to the comment on lines 513-530, we believe it is essential to focus on a singular discussion topic: “Dispersal Mechanisms and Species Distribution: Impacts of Primary and Secondary Dispersal Vectors on the Invasive Potential of Lepidium draba and Lepidium campestre.” In the introduction, we have highlighted the native and total distribution areas of both invasive species. While we have provided these details, we recognize the need for a more comprehensive comparison of their dispersal mechanisms and distribution areas in the discussion section. Therefore, we structured this section around the aforementioned topic to ensure that these aspects are thoroughly addressed. We hope this explanation clarifies our approach and meets the reviewer’s expectations.
- Line 86. Change “of the two species “ to “L. campestre” Your figure 2 shows mucilage only for L. campestre.
Please be aware that both species produces mucilage. Even in the introduction section we mentioned that one of the common characteristics of the two species is that both produces mucilage. We have cited our paper for this (
https://doi.org/10.1016/j.actao.2024.104042). However, we would like to clarify that our current study focuses specifically on primary dispersal. For Lepidium draba, the whole fruit serves as the true dispersal unit rather than isolated seeds. Since the seeds remain enclosed within the fruit and do not contribute to primary dispersal, we deemed it unnecessary to present data on mucilaginous seeds for this species. Note that the true dispersal unit of Lepidium campestre is isolated seeds. We hope the reviewer follows our point.
- Line 103. Change ”(L.)” to “L.”
The bracket is needed here, but we have added a dot. Therefore, the corrected version is (L.).
- Line 105. Insert “were” before “collected”
The correction has been made as requested.
- Line 106. Delete “propagated and”
We have removed the phrase "were mass propagated" to enhance clarity and improve the overall understanding of the text.
- Line 108. Not clear what you did with the transgenic Lepidium campestre seeds. Did you do experiments with them?
Thank you for your question regarding the transgenic Lepidium campestre seeds. We would like to clarify that we did conduct experiments using the fruits from the transgenic Lepidium campestre. Specifically, these experiments are detailed in Figure 6 of the manuscript. In our study, we compared the puncture force of wild Lepidium campestre fruits (which produce dehiscent fruits), transgenic Lepidium campestre fruits (which produce indehiscent fruits), and Lepidium draba fruits (also indehiscent). We hope this clarification addresses your query effectively.
- Line 113. Delete “method”
We have made the necessary corrections as requested.
- Line 114. After “infructescences” should you insert “of both species” Of, did you only do the raindrop test with one of the species? Not clear
Thank you for your insightful question. We have added the phrase “of both species” for clarity. To clarify further, the experiments were indeed conducted by comparing both species. In the methods section, we specified that we recorded the number of individual raindrops, with a mean mass of 53.55 ± 0.97 mg, impacting the fruit/seed until either fruit abscission (in dehiscent and indehiscent fruits) or seed detachment (in M+ seeds) occurred, following the methodology described by Arshad et al. (2019). This should make it clear that Lepidium campestre produces dehiscent fruits while Lepidium draba produces indehiscent fruits. We believe that the addition we made will enhance the understanding of this point.
- Line 137. Change “were” to “was” maximum was
The corrections have been made as requested.
- Line s156-157. Delete the last sentence of the paragraph. This is something you might say in the Discussion.
The last sentence of the paragraph has been deleted as suggested.
- Line 164. Need to say what the different types of diaspore of campestre are.
We have added the following clarification: “dry mucilaginous seeds (dry M+), wet mucilaginous seeds (wet M+), and re-dried mucilaginous seeds (re-dried M+ seeds).”
- Line 172. Insert a comma before “while”
The corrections have been made, and the entire paragraph has been revised. As a result, the word "while" was removed. Please see the modified version of the paragraph for your review.
- Line 175. In this section of the methods you do not tell the reader how you studies after-ripening of the seeds. However, you talk about after-ripening in the Results. Need to provide the methods for the after-ripening studies.
Thank you for your valuable feedback. The reviewer raised an important point, and we have addressed it by adding the following information: “To assess the impact of after-ripening on mechanical puncture force measurements, freshly harvested fruits were stored under controlled laboratory conditions (25 ± 2 °C, 51% relative humidity) for 16 weeks, following the method described by Mohammed et al. (2019).”
- Line 184. I found it very difficult to follow and understand your Results. It seems that you have not presented the Results in the same order that you present the Methods. You need to either put the methods or the results in a different order, that is they need to be in the same order.
Dear reviewer, thank you for your valuable feedback regarding the organization of the Results section. We understand the importance of aligning the Methods and Results for clarity. In response to your comments, we have reorganized the subtopics in the Methods section to match the order presented in the Results section. Specifically, we have moved the wire-hook induced force applied to detach seeds and fruits from the infructescence to the beginning of the Methods section. We believe that this change enhances the overall coherence and readability of the manuscript.
- Lines 186-196. Since you talk about raindrops in the first section of the methods, you should be giving the results from the raindrop test here.
You are absolutely correct that it would improve clarity to present these results immediately following the relevant methods. In response to your comments, we have restructured the Methods section to ensure that the results from the raindrop test are discussed in conjunction with the appropriate methods, as previously mentioned in our response to question #17. This adjustment has significantly improved the flow and readability of the manuscript. We appreciate your insights and believe these changes enhance the overall quality of our work.
- Line 187. What force are you talking about? Raindrops falling? Not clear – heard to follow/understand.
We would like to clarify that the details you seek are explicitly stated in Figure 3B. Small wire hooks were placed at the seed and replum junction, and the mass of these hooks required to detach the seeds from the replum was used to calculate the force (N) needed for detachment. You can find additional information in the figure legend. We believe that the methods section, in conjunction with the figures, provides clear and sufficient information regarding this process.
- Line 187. Delete “was observed”
The corrections have been made successfully.
- Line 190. Delete “the results indicated that”
We have deleted the phrase “The results indicated that” and revised the sentence to begin with an uppercase letter as follows: “There was….”
- Line 195. What force? Is this section about the results of puncture test?
This section referred to the wire-hook induced force applied to detach seeds and fruits from the infructescence. The material and methods section has been altered to clarify this. For the wire-hook experiments, please refer to the method outlined in lines 159-166 of the manuscript: Small-sized wire hooks were placed at the seed and replum junction, and the mass of the wire hooks required to detach the seeds from the replum was used to calculate the force (N) necessary for this detachment, utilizing the formula: Force (N) = mass (in kg) multiplied by acceleration (m/s²) (see also Arshad et al., 2019). A total of 200 samples for each type of diaspore from Lepidium campestre were tested, yielding a robust dataset for analysis. The aim of this study was to enhance our understanding of the mechanical dynamics affecting diaspore dispersal in Lepidium campestre. Any further biomechanical analysis is now described in the “Biomechanical analysis” section of the Material and Methods.
- Lines 223-230. Now, I find the results from the raindrop test. I strongly recommend that you arrange the topics in the methods and in the results in the same order. A reader should be able to read the first section of the methods and then turn to the first section of the results and learn what happened during the first experiment. Now, your results are a big mixture of topics that is not in the same order as the topics in the methods. Please make the reading of your interesting manuscript an easy experience for the reader.
We completely understand your concern and appreciate your suggestion for a more coherent structure. In line with your recommendations and our previous responses to questions #17 and #18, we have thoroughly rearranged the Method section to mirror the order of the Results section. This adjustment greatly enhances the readability and flow of the manuscript, making it easier for readers to follow the progression of the experiments. We are grateful for your input, which has been invaluable in improving our manuscript.
- Line 232. Delete “it was found that”
The corrections have been made successfully.
- Line 311. Delete “the study reported that”
We have deleted …“The study reported that” and the sentence started with uppercase letter The …..
- Line 336. Delete “The results showed that”
We have removed the phrase "The results showed that" as per your suggestion and the sentence started with uppercase letter “None”….
- Line 370. Near the first of the Discussion, you need to tell the reader if your hypothesis was support (or not) and talk about the data that support (or not) your hypothesis.
We appreciate your suggestion, and we recognize the importance of clearly articulating the support for our hypothesis within the discussion section. In our manuscript, we prioritize introducing essential concepts, such as primary and secondary dispersal vectors, at the beginning of the discussion and the evolutionary relationship between dehiscent and indehiscent fruit producing Lepidium species. This foundational context is crucial for readers to fully understand the subsequent analysis of the ecological and evolutionary implications of our findings. While we agree that addressing our hypothesis early on is important, we believe it should be integrated into the context of the discussion rather than introduced first, as this can vary based on the research topic. Nonetheless, we will ensure that the discussion section clearly addresses whether we accept or reject our null hypothesis, thus enhancing the overall clarity and logical flow of the manuscript.
- Lines 474-475. Your need to provide the methods for after-ripening.
We recognize the importance of this information and have addressed it in our response to question #16.
- Line 554, change “influenced” to “correlated”
We appreciate your observation regarding the terminology. We have replaced the word "influenced" with "correlated" to better reflect the nature of the relationship discussed in the manuscript. Consequently, this change required us to adjust the subsequent wording; therefore, we substituted "by" with "with" to maintain grammatical accuracy.
Reviewer 4 Report
Comments and Suggestions for Authors
The manuscript entitled: ‘Differential Primary Seed and Fruit Dispersal Mechanisms and Puncture Force Measurements in Invasive Dehiscent and Indehiscent-Fruited Lepidium Species’ is a comprehensive research on the the reproductive strategies of plants. In my opinion, such a detailed research will certainly be of great interest to researchers around the world as the study addresses the highly important topic of the dispersal mechanisms of invasive plants. The manuscript is suitable for publication in the journal after major revision.
The title represents well the content of the manuscript.
The quality of English does not limit my understanding of the research.
The Materials and Methods was not presented in a clear manner. The measurement methods should be separated from ‘2.1. Seed Sources’, such as: 2.2. Puncture Force Measurements, 2.2.1. Fruit, fruit valve, and seed resistance to raindrop impact, 2.2.2. Simulated rainfall experiment to assess seed and fruit dispersal......
The result part of the manuscript displayed several Puncture Force Measurement results of Lepidium fruits. However, only these results were not enough. Authors should to explain why there are divergent dispersal strategies of Lepidium Species. I encourage Authors to add more data, such as the ultramicrostructures of the seed-pedicel junction, seed-replum junction and fruit-pedicel junction, as well as the application of multi-omics technology.
In Discussion, the comparison with other findings deserves special attention.
Author Response
Detailed response to the Reviewer
Reviewer #4
Comments and Suggestions for Authors
The manuscript entitled: ‘Differential Primary Seed and Fruit Dispersal Mechanisms and Puncture Force Measurements in Invasive Dehiscent and Indehiscent-Fruited Lepidium Species’ is a comprehensive research on the the reproductive strategies of plants. In my opinion, such a detailed research will certainly be of great interest to researchers around the world as the study addresses the highly important topic of the dispersal mechanisms of invasive plants. The manuscript is suitable for publication in the journal after major revision.
The title represents well the content of the manuscript.
The quality of English does not limit my understanding of the research.
- The Materials and Methods was not presented in a clear manner. The measurement methods should be separated from ‘2.1. Seed Sources’, such as: 2.2. Puncture Force Measurements, 2.2.1. Fruit, fruit valve, and seed resistance to raindrop impact, 2.2.2. Simulated rainfall experiment to assess seed and fruit dispersal......
We have revised the Material and Methods as you suggested and according to the proposal of reviewer #3 point 17. We hope this improved the Material and Method section.
- The result part of the manuscript displayed several Puncture Force Measurement results of Lepidium fruits. However, only these results were not enough. Authors should to explain why there are divergent dispersal strategies of Lepidium Species. I encourage Authors to add more data, such as the ultramicrostructures of the seed-pedicel junction, seed-replum junction and fruit-pedicel junction, as well as the application of multi-omics technology.
We want to clarify that our study includes a variety of experiments beyond the biomechanical measurements presented in Figure 6. In addition to the puncture force experiment, we conducted several other experiments, including the assessment of fruit, fruit valve, and seed resistance to raindrop impact; a simulated rainfall experiment to evaluate seed and fruit dispersal; sheep wool contact experiments to simulate diaspores attachment and detachment from the infructescence; wind-induced dispersal of diaspores; and wire-hook induced force applications to detach seeds and fruits from the infructescence. Thus, the puncture force measurement is just one component of our comprehensive investigation. From our research objectives and the scope of the project, we believe that conducting additional experiments on ultramicrostructures is not necessary for our current focus. We acknowledge that individual studies can address specific questions effectively, and we are confident that our data on the primary dispersal mechanisms of the two invasive species provides a complete narrative on their dispersal strategies. Moreover, we have plans to apply for a future project that will include topics related to multi-omics technology. We appreciate your understanding of the context of our study, and we hope that our methods and results convey a complete story that is relevant to our research aims.
- In Discussion, the comparison with other findings deserves special attention.
Thank you for your valuable feedback regarding the need for comparisons with other findings in the Discussion section. We made an effort to correlate our results with existing literature of course we did; however, it is important to note that our study specifically addresses invasive species (weeds) and focuses on the primary dispersal strategies of two related Lepidium species. The limited availability of literature on this niche topic poses significant challenges. As experts in invasive Lepidium species, we are familiar with the key studies in this area. While we recognize that our manuscript may lack broader comparisons, this is largely due to the specificity of our research focus rather than an oversight. We hope this explanation clarifies the context of our work and the challenges we faced in finding relevant comparisons. We appreciate your understanding and hope that it provides a satisfactory response to your suggestion.
Round 2
Reviewer 2 Report
Comments and Suggestions for Authors
This manuscript has been drastically improved, and is very much better. I think it will be of great interest to those interested in weeds and their dispersal, especially interesting since these two Lepidium spp. are native to different parts of the world.
I have only a few small corrections:
Line 66 – I think the name of the mutuation should be spelled INDEHISCENT…?
Line 511 – de-capitalize specific epithets in the heading, and de-italicize latin names in an italicized sentence
Line 550 – de-itaclicize latin name in heading (and throughout in italicized sentences)
Author Response
Detailed response to reviewer # 2, round 2
Comments and Suggestions for Authors
This manuscript has been drastically improved, and is very much better. I think it will be of great interest to those interested in weeds and their dispersal, especially interesting since these two Lepidium spp. are native to different parts of the world.
I have only a few small corrections:
- Line 66 – I think the name of the mutuation should be spelled INDEHISCENT…?
Changes made as requested.
- Line 511 – de-capitalize specific epithets in the heading, and de-italicize latin names in an italicized sentence
Changes made as requested
- Line 550 – de-itaclicize latin name in heading (and throughout in italicized sentences)
Changes made as requested

Reviewer 3 Report
Comments and Suggestions for Authors
Line 35. Change “Biomechanics” to “biomechanics”
Line 42. Change “facilitate” to “facilitates”
Line 67. Change ‘which” to “that”
Line 99. Change “producing” to “of”
Line 123. Change “(L.)” to “L.”
Line 206. Change “was” to “were”
Line 235. Change “detaching” to “are detached”
Line 252. Delete “(Lc)”
Line 3313. Change “required detaching” to “required to detach”
Line 358. Change “through” to “in”
Line 429. Insert “to” before “newly”
Line 429. Delete “have”
Line 438. Change to “indehiscent fruits could be associated…”
Line 442. Change to “Primary dispersal is defined as the initial dispersal of seeds and fruits from the mother plants…..”
Line 444-445. Change to “referred to as secondary dispersal….”
Line 446. Change “as” to “since”
Line 448. Insert “was” before “associated”
Line 455. Insert “to be” after “likely”
Line 461. Change “involves” to “involve”
Line 469. Change “upon” to “after”
Line 469. Change “being” to “is”
Line 483. Change “This” to “These”
Line 491. Change “as” to “since”
Line 517. Change “on” to “through”
Line 557. Change “This indicates that” to “Thus,”
Line 590. Change “releases” to "is released”
Author Response
Detailed response to reviewer # 3, round 2
Comments and Suggestions for Authors
- Line 35. Change “Biomechanics” to “biomechanics”
Changes made as requested.
- Line 42. Change “facilitate” to “facilitates”
Changes made as requested.
- Line 67. Change ‘which” to “that”
Changes made as requested.
- Line 99. Change “producing” to “of”
Changes made as requested.
- Line 123. Change “(L.)” to “L.”
Changes made as requested.
- Line 206. Change “was” to “were”
Changes made as requested.
- Line 235. Change “detaching” to “are detached”
Changes made as requested.
- Line 252. Delete “(Lc)”
Instead of deleting the abbreviation 'Lc', it has been changed to the full name, L. campestre.
- Line 3313. Change “required detaching” to “required to detach”
Changes made as requested.
- Line 358. Change “through” to “in”
Changes made as requested.
- Line 429. Insert “to” before “newly”
Changes made as requested.
- Line 429. Delete “have”
Changes made as requested.
- Line 438. Change to “indehiscent fruits could be associated…”
Changes made as requested.
- Line 442. Change to “Primary dispersal is defined as the initial dispersal of seeds and fruits from the mother plants…..”
Changes made as requested.
- Line 444-445. Change to “referred to as secondary dispersal….”
Changes made as requested.
- Line 446. Change “as” to “since”
Changes made as requested.
- Line 448. Insert “was” before “associated”
Changes made as requested.
- Line 455. Insert “to be” after “likely”
Changes made as requested.
- Line 461. Change “involves” to “involve”
Changes made as requested.
- Line 469. Change “upon” to “after”
Changes made as requested.
- Line 469. Change “being” to “is”
Changes made as requested.
- Line 483. Change “This” to “These”
Changes made as requested.
- Line 491. Change “as” to “since”
Changes made as requested.
- Line 517. Change “on” to “through”
Changes made as requested.
- Line 557. Change “This indicates that” to “Thus,”
Changes made as requested.
- Line 590. Change “releases” to "is released”
Changes made as requested.
